# Origins of East Caucasus Gene Pool: Contributions of Autochthonous Bronze Age Populations and Migrations from West Asia Estimated from Y-Chromosome Data

**DOI:** 10.3390/genes14091780

**Published:** 2023-09-09

**Authors:** Anastasia Agdzhoyan, Nasib Iskandarov, Georgy Ponomarev, Vladimir Pylev, Sergey Koshel, Vugar Salaev, Elvira Pocheshkhova, Zhaneta Kagazezheva, Elena Balanovska

**Affiliations:** 1Research Centre for Medical Genetics, 115522 Moscow, Russiafreetrust@yandex.ru (V.P.); eapocheshkhova@mail.ru (E.P.);; 2Biobank of Northern Eurasia, 115201 Moscow, Russia; 3Department of Cartography and Geoinformatics, Faculty of Geography, Lomonosov Moscow State University, 119991 Moscow, Russia; 4Department of Biology with Course in Medical Genetics, Faculty of Pharmacy, Kuban State Medical University, 350063 Krasnodar, Russia

**Keywords:** East Caucasus, gene pool, Y-chromosome, Bronze Age populations, Dagestan, Azerbaijan, migrations, West Asia

## Abstract

The gene pool of the East Caucasus, encompassing modern-day Azerbaijan and Dagestan populations, was studied alongside adjacent populations using 83 Y-chromosome SNP markers. The analysis of genetic distances among 18 populations (*N* = 2216) representing Nakh-Dagestani, Altaic, and Indo-European language families revealed the presence of three components (Steppe, Iranian, and Dagestani) that emerged in different historical periods. The Steppe component occurs only in Karanogais, indicating a recent medieval migration of Turkic-speaking nomads from the Eurasian steppe. The Iranian component is observed in Azerbaijanis, Dagestani Tabasarans, and all Iranian-speaking peoples of the Caucasus. The Dagestani component predominates in Dagestani-speaking populations, except for Tabasarans, and in Turkic-speaking Kumyks. Each component is associated with distinct Y-chromosome haplogroup complexes: the Steppe includes C-M217, N-LLY22g, R1b-M73, and R1a-M198; the Iranian includes J2-M172(×M67, M12) and R1b-M269; the Dagestani includes J1-Y3495 lineages. We propose J1-Y3495 haplogroup’s most common lineage originated in an autochthonous ancestral population in central Dagestan and splits up ~6 kya into J1-ZS3114 (Dargins, Laks, Lezgi-speaking populations) and J1-CTS1460 (Avar-Andi-Tsez linguistic group). Based on the archeological finds and DNA data, the analysis of J1-Y3495 phylogeography suggests the growth of the population in the territory of modern-day Dagestan that started in the Bronze Age, its further dispersal, and the microevolution of the diverged population.

## 1. Introduction

The East Caucasus, which spans modern-day Dagestan and Azerbaijan, is an important land bridge connecting Europe to West Asia. One of the gateways through the Caucasus is the Big Caucasian Pass, a strip of coastal land that runs between the East Caucasus mountains and the Caspian Sea all the way from Derbent to Sumgait.

The early humans of the Oldowan culture dispersed to the East Caucasus about 2 million years ago [1]. The ancient Chokh settlement site in today’s Dagestan is the key monument of the Mesolithic, Neolithic, and Bronze Age cultures that used to exist in the mountainous areas of the Northeast Caucasus. Evidence from recent radiocarbon dating suggests that domestication of animals, crop cultivation, and pottery making in this region began no later than the late 7th or early 6th millenniums BC [2]. Agriculture was booming; according to Nikolai Vavilov, the mountainous parts of Dagestan were the center of terraced farming [3]. The economic development during the Neolithic period (crop farming and animal husbandry, the advent of pottery, weaving, stone grinding, and polishing techniques) led to rapid population growth and colonization of new territories. In Azerbaijan, advanced Neolithic societies emerged at the dawn of the 6th millennium BC as a succession from the Neolithic societies of Southwest Asia [4,5]. The discovery of metals in the Eneolithic period (~7 kya) spurred the development of technology. The Bronze Age (6–4 kya) was marked by the emergence of new metalwork techniques, economic and social changes (social inequality, tribal confederations), and trade relations with the populations of Southeastern Europe and West Asia. In the Middle Bronze Age, nomads from the Eurasian steppe arrived in the East Caucasus, introducing the kurgan burial custom into the region.

In the 7th–6th centuries BC, the south of the region was infiltrated via the Derbent Pass by the Iranian-speaking Scythians who formed a military-political alliance known as Ashguza on the lands of today’s West Azerbaijan and Northwest Iran. Later, this territory was annexed by the Median Kingdom. In the late 1st millennium BC, the state of Caucasian Albania emerged from a tribal confederation in the northern part of the region. It had its own writing system, and its population adopted Christianity in the 4th century AD. Caucasian Albania was largely made up of autochthonous Caucasian tribes that spoke the Lezgic languages of the Nakh-Dagestanian family, although its reign extended over some Iranian tribes too. The territory of modern-day Dagestan was a cradle to such early states as Lakz, Tabarseran, Haidak, Sarir, Gumik, etc., which came into being in the first half of the 1st millennium AD.

During the first few centuries AD, other Iranian tribes, including Sarmatians, Maskut, and Alans, were arriving in the Caucasus. The South Caucasus was raided by Turkic-speaking Huns, Sabirs, and Khazars who came from the North via the Derbent Pass in the early Middle Ages (between the late 4th and 8th centuries) [6]. In the 7th century AD, the north of lowland Dagestan was occupied by the rising Khazar Khanate, whereas the south of the region was colonized by the Sasanian Empire in the 4th century. In the 7th–9th centuries, the East Caucasus was controlled by the Arab Khalifate, which was pursuing an aggressive migration policy that resulted in the Islamization of the local population. Incursions of Turkic-speaking Seljuks, the founders of the powerful empire spanning Central and West Asia and the South Caucasus, began in the 11th century as the invaders continued their expansion from Central Asia into the East Caucasus, including Derbent. Turkic peoples from the Cumania (Desht-i-Kipchak) migrated to the South Caucasus in the 11th–12th centuries; their descendants may have contributed to the emergence of the subethnical Azebaijani-Karapapakh group [7]. Other Turkic-speaking peoples of the East Caucasus are Kumyks and Nogais. There is no consensus on the origin of Kumyks. The prevailing hypothesis traces their descent to the local autochthonous population that had close ethnocultural contacts with foreign Turkic tribes like Sabir, Khazar, or Kipchak. In turn, Nogais were nomads from a late migration wave; most of them are settled in Dagestan and are represented by the Karanogai population.

After the Tatar-Mongol expansion that started in the 14th century, Islamic influence regained its primacy in Dagestan. The local medieval principalities of that time existed until the 19th century when they were absorbed by the Russian Empire. The southern regions of the East Caucasus were under Iranian influence during the 15th–19th centuries; they were incorporated by the Russian Empire in the aftermath of the Russo-Persian wars when the peace treaties were signed in 1813 and 1828. Later, these territories became the state of Azerbaijan.

Thus, the gene pools of the autochthonous East Caucasus populations were influenced by the massive ancient migration waves of Iranian speakers and later migrations of Eurasian Turkic peoples. The rich history of the region is reflected in the linguistic diversity of the East Caucasus. The languages of the region include the Nakh-Dagestanian languages (spoken by over 30 ethnic and ethnographic groups), the Turkic languages of the Altaic language family (spoken by Azerbaijanis, Kumyks, and Karanogais), and the Iranian languages of the Indo-European language family (spoken by Tats, mountain Jews, Yezidis, Kurds, and the Talysh) [8]. This stunning diversity of Eastern Caucasian populations is a major obstacle in the study of the East Caucasus gene pool, even if highly effective genogeographic tools are used, such as genotyping for Y-chromosome markers. 

Indeed, there is a wealth of publications describing the gene pools of East Caucasian populations [9,10,11,12,13,14,15,16,17,18,19,20,21,22,23,24,25,26]. However, genotyping panels have expanded dramatically in the past 20 years and now include new informative markers. For example, approximately 13,000 Y-SNPs were discovered using the NGS technology in 2015; of them, 66% had been previously unknown [27]). The results of early genetic studies were not so informative and cannot be compared to the data generated with the help of modern genotyping panels. 

Samples of Azerbaijani populations were studied for Y-chromosome markers by [9,10,11,12,16]. Some population studies point to the similarity between the gene pools of Azerbaijanis and their neighbors in the Caucasus and West Asia [11]. Azerbaijanis have genetic affinity to the Iranian/Turkic-speaking populations of West Asia but differ from Turkmens [16]. According to [20,21], Iranian Azerbaijanis bear genetic similarities to their Iranian-speaking neighbors, while the Azerbaijani-Terekeme of Dagestan have a genetic affinity to Kumyks [23]. Two more publications [25,26] report the results of commercial DNA testing among the Azerbaijanis. Briefly, the specimens were obtained from volunteers who were willing to learn about their Y-chromosome lineage and could afford the test. The results of the test revealed the prevalence of West Asian lineages in the Azerbaijani gene pool; however, the accumulated published data suggests that 18% of their gene pool reflects medieval migrations from Central Asia and 6% is associated with migrations from East Asia [25]. 

The Azerbaijani gene pool remains understudied due to the use of small genotyping panels or small samples [9,10,11,12], or study populations from neighboring regions—Dagestan or Iran [16,20,21,23]. It might be difficult to use the data generated by such research at the current level of Y-haplogroup resolution and to comprehensively describe the Azerbaijani gene pool. The data obtained by our colleagues is intended for the analysis of broader subjects than the genetic composition of the East Caucasus. However, the absence of a substantial Cen-tral Asian component and the genetic similarity of Azerbaijanis to the populations of the Middle East and Caucasus give some general idea about their gene pool.

Dagestani populations have been studied for Y-chromosome markers much better [13,14,17,18,22,24,28] than the populations of Azerbaijan. For example, a significant correlation was found between the frequencies of Dagestani haplogroups, their geography and the language (as demonstrated by the lexicostatistical data) [17]. A similar finding was reported by Karafet T. [22], who pointed to the high interpopulation diversity of Dagestan’s indigenous peoples and confirmed the hypothesis about the descent of all Nakh-Dagestanian speaking populations from a single protopopulation ~6000–6500 years ago. The descent of Dagestani-speaking peoples from a single postglacial ancestral population and the subsequent genetic divergence due to landscape changes, genetic drift, and the founder effect were described in [14]. Unlike the mountain people of Dagestan, its lowland populations bear a greater Y-chromosome diversity, which can be explained by their contacts with Central Asian populations [13]. The Tsez, who represent the Avar-Andi-Dido language family, demonstrate a modest genetic diversity due to the prevalence of the JI haplogroup and the strong founder effect [24]. Dagestan is one of the most ethnically diverse Russian regions; its gene pool was system-atically analyzed in the past using the genotyping panels of Y-chromosome markers available at that time [17,18,22], and the results of the analysis are now outdated. It is known that the J1 haplogroup occurs in the populations of Dagestan at high frequency, so it is important to explore its structure and the distribution of its branches in order to fur-ther analyze the history of Dagestan’s populations.

Thus, the aim of this study was to systematically analyze the gene pools of popula-tions inhabiting the historically connected territories of Dagestan and Azerbaijan and to reconstruct the genetic history of the East Caucasus using a broad panel of Y-chromosome markers. The results of this study could be used to model the dispersal of early humans across the Caucasus, West Asia and Eurasia, using the updated and expanded DNA data.

## 2. Materials and Methods

*Samples*. We have analyzed 18 populations from the East Caucasus and the adjacent regions (N = 2216; Table 1) that represent 3 language families: Nakh-Dagestanian (the Avar-Andi-Dido, Dargin, Lak, Lezgi, and Nakh branches), Altaic (the Turkic branch) and Indo-European (the Iranian branch). the focus of the study was on the East Caucasus, the gene pools of the indigenous peoples of Dagestan and Azerbaijan (N = 1329) were analyzed in greater detail than North Caspian steppe populations, Nakh and Iranian speakers of the Caucasus, which constituted the comparison group and were represented by pooled samples without further division into smaller subgroups. The Nakh-Dagestanian language family was represented by 14 ethnic groups: Avars and Tindi (the Avar-Andic language branch), Dido and Hinukh (the Tsez branch), Dargins, Kaitags and Kubachi (the Dargin branch), Laks (the Lak branch), Lezgins, Rutuls, Tabasarans and Tsakhur (the Dargin branch), and Chechens and Ingush (the Nakh branch). Turkic speakers were represented by the populations of the East Caucasus (Karanogais, Kumyks, the Azerbaijani Terekeme of Dagestan, and the Azerbaijanis of Azerbaijan) and the West of the Eurasian steppe (Stavropol Nogais, Trukhmens, Astrakhan Tatars and Nogais). Iranian-speaking populations of the Caucasus were represented by Tats, Mountain Jews, Yezidis, Kurds, and Talysh. More than half of the studied samples are published here for the first time. The samples that had been studied in ([17,23], Appendix A) were additionally genotyped using an expanded SNP panel. 

The biological specimens were collected during the 1998–2018 expeditions under the supervision of Prof. Elena Balanovska, Prof. Oleg Balanovsky, and Prof. Elvira Pocheshkhova. The Azerbaijani population was studied back in 2015–2022; that research initiative was supported by the Azerbaijani expat community. All samples were collected from unrelated male donors whose ancestors from at least 3 previous generations belonged to the studied ethnicity and population. Informed consent was obtained from all donors. The study was approved by the Ethics Committee of the Research Centre for Medical Genetics.

*Molecular Genetic Testing*. In the preparatory step, DNA was isolated from the samples of venous blood and saliva using a QIAsymphony SP instrument or by phenol-chloroform extraction. DNA concentrations were measured with a Nanodrop 2000 spectrophotometer and a Qubit 4.0 fluorometer. The samples were aliquoted and distributed into well plates for further genotyping using a QIAgility workstation.

The samples were genotyped using OpenArray technology, a QuantStudio 12 Flex Real-Time PCR system, TaqMan probes, and a custom panel of the most informative SNP markers. The set of 83 studied SNPs included two subsets: the “Core” array of 62 SNPs and a J1-M267(×P58) array of 31 SNPs (Appendix A). The “Core” array covered the main haplogroups of the world’s Y-chromosome tree, specific to the indigenous populations of Northern Eurasia, which spans Russia, post-Soviet states and Mongolia. Since the J1-M257(×P58) haplogroup has a pronounced presence in the gene pool of the Eastern Caucasus [17,18], the J1-M257(x(×P58) array was used to study its phylogeography. The dates of the haplogroups’ origin provided in this article were borrowed from YFull [https://www.yfull.com/, accessed on 31 May 2023] if not specified otherwise.

*Statistical analysis*. The pairwise matrix of Nei’s genetic distances [29] (Table 2) was computed in the original DJ software [30] from the frequencies of 38 Y-chromosome haplogroups that are polymorphic in the studied populations (Appendix A). The visual plot for the matrix was created in Statistica 7.0 (StatSoft©, Tulsa, Oklahoma, United States) using the multidimensional scaling method.

*Cartographic analysis*. Genogeographic maps (frequency distribution maps, maps of genetic distances) were constructed from the genotyping data and the information from the Y-base database (Table 1 and Appendix A), which had been developed under the supervision of Prof. Oleg Balanovsky. The original GeneGeo software used to create the genogeographic maps had been developed under the supervision of Prof. Elena Balanovska and Prof. Oleg Balanovsky. Frequency distribution maps for the studied Y-chromosome haplogroups were created using the average weighted interpolation procedure with a search radius of 400 km and a weight function inversely proportional to the cube of the distance [31]. The maps of genetic distances were constructed for 39 Y-chromosome haplogroups of the studied East Caucasus populations using an algorithm similar to the one described in [32,33,34].

## 3. Results and Discussion

### 3.1. The Spectrum of Y-Chromosome Haplogroups in the Populations of the East Caucasus and the Adjacent Regions

While studying the genotyping data of 18 indigenous populations of the East Caucasus and the adjacent regions (N = 2216), we factored their language and area into the analysis. The geographic variability of the analyzed Y-chromosome haplogroups across the East Caucasus (Figure 1, Appendix A) is not the same for the representatives of different language families: it is more pronounced for Nakh-Dagestanian linguistic groups than for Turkic and Iranian-speaking populations.

Overall, we have identified 5 major haplogroups with different geographic distribution patterns: J1, J2, R1a, R1b, and N. Their diversity is demonstrated on the frequency distribution maps (Figure 2). The distribution of haplogroup J1 follows a distinct focal pattern: in Dagestan, its frequency declines in all directions from the center to the periphery (Figure 2D). Haplogroup J2-M172(×M67, M12) also has a focal pattern, but its peak frequencies occur in the south of the region: the cumulative contribution and diversity of J2 branches is greater for the Azerbaijanis and the Iranian-speaking populations of the Caucasus (Figure 2E–G). R1b-M269 echoes this pattern, occurring at high frequencies in the south of the region among the Azerbaijani Talysh (48% [35]; Figure 2E–H). Overall, haplogroups R1a and R1b expose the connection between the steppe populations in the north and the groups settled along the Caspian coastal line in the south.

Note. The red dots on the map represent the analyzed populations. Areas of grey indicate haplogroup frequencies below 1%, areas of green indicate low frequencies of 1–20%, areas of yellow are the zones of moderate frequencies (20–50%), and areas of red and purple are zones of high frequencies (>50%). 

Legend:

1—Karanogais, 2—Chechen, 3—Kumyks, 4—Tindi, 5—Avars, 6—Dargins, 7—Tsez (Dido) and Hinukh, 8—Laks, 9—Kaitak, 10—Azerbaijanis from Dagestan, 11—Kubachi, 12—Tabasarans, 13—Tsakhur, 14—Rutuls, 15—Lezgins, 16—Karapapakhs, 17—Azerbaijanis (including Karapapakhs) from Azerbaijan, 18—Azerbaijanis (excluding Karapapakhs) from Azerbaijan, 19—Iranian-speaking peoples of the Caucasus (Tats, Talysh, Kurds), 20—Iranian-speaking peoples of the Caucasus (Yezidis).

Haplogroup J1 dates back 9000 to 24,000 years [36] and is thought to have originated in West Asia [37,38,39], where its J1-P58 branch is the most prevalent and reaches the peak frequency in the Arabs of the Middle East [36]. The J1-P58 lineage is extremely rare for the East Caucasus populations; in this region, haplogroup J1 is represented by the J1-M267(×P58) clade that accounts for over 50% of the gene pool of Dagestani-speaking populations. This finding is consistent with other publications [14,17,18,19,22,24]. The two exceptions are the Tsakhur (44%) and Tabasaran (13%) peoples. The frequency of J1-M267(×P58) culminates in the Kubachi (98%), whose habitat is confined to one village. J1-M267(×P58) prevails in other Dagestani populations (59–92%) and is frequent among the Chechens (17–25%). However, its frequency drops dramatically as the distance from the Nakh-Dagestanian speakers grows (Figure 1, Appendix A). 

The high frequency of haplogroup J1-M267(×P58) observed for Kumyks may indicate a significant genetic contribution of Nakh-Dagestanian-speaking populations since this haplogroup does not typically occur in other Turkic and Iranian populations of the East Caucasus (Figure 1).

Apart from the genetic drift, the presence of other J1 branches in addition to J1-P58 in the region may be responsible for the high frequency of J1-M267(×P58). Since the early carriers of haplogroup J1 emerged in the Caucasus as early as the Paleolithic (~13 kya; hunter-gatherers in the territory of modern-day Georgia [36,40]) and the rise of farming communities in the Neolithic drove population growth and the emergence of new haplogroups, a few local lineages may have arisen within J1-M267(×P58). This hypothesis will be explored below. 

Haplogroup J2 is rarely found in the populations of Dagestan but its J2-M67(×M92) branch occurs at 67% frequency among their neighbors, the Nakh-speaking Chechens and Ingush (Figure 1 and Figure 2F, Appendix A). J2 is represented at 20% frequency by its clade J2-M172(×M67, M12) in the gene pools of the Azerbaijanis and Iranian-speaking peoples of the Caucasus. 

The origin of haplogroup J2 is traced to West Asia [38,41,42], where the diversity of its branches is the greatest (J2a-M92 and J2a- M530 [20]); its frequencies culminate in Iranian-speaking populations. In West Asia, the main carriers of haplogroup J2 are Anatolian Turks, Azerbaijanis, Kurds, and Persians (20–35%); the most prevalent is the J2-M172(×M67, M12) branch with 15% frequency in Turkic and 20% frequency in Iranian-speaking populations (Figure 2A). In the East Caucasian gene pool, its frequency is 11% for Turkic speakers and 17% for Iranian speakers (Appendix A).

Thus, the frequencies of haplogroup J2 and its branch J2-M172(×M67, M12) among the Azerbaijani-Terekeme of Dagestan and the Azerbaijanis of Azerbaijan are close to the frequencies observed not only for the Iranian-speaking peoples of the Caucasus but also for the populations of West Asia.

Haplogroup N reaches its highest frequencies in the steppe population of Karanogais in the North of Dagestan (22%) and Azerbaijani Karapapakhs (10%, Figure 1, Appendix A). However, its lineages differ between the two populations: N3a5a-F4205, and N3a2-M2118, both of which are common in Siberia and Central Asia, and the undifferentiated branch N-LLY22g(×M178, Y3205) occur in the Karanogais, whereas N2-Y3205 and N3a1-B211, which are typical for the populations of the Ural region and West Siberia, are found in the Karapapakhs. 

The cumulative contribution of haplogroups R1a and R1b is the most substantial for Dagestani Turkic peoples (24–40%), Azerbaijanis of Azerbaijan (30%), and Iranian-speaking populations of the Caucasus (37%). Its frequencies decline smoothly in the Nakh-Dagestanian populations of Dagestan except for the Tabasaran (Figure 1, Appendix A). 

Haplogroup R1b is represented by 2 subclades: R1b-M269 and R1b-M73, of which the former is more common. The peak frequency of R1b-M269 is observed for the Tabasaran (64% in our study, 10% in [14] and 40% in [18]), although it does not exceed 15% for most Nakh-Dagestanian linguistic groups (Appendix A). R1b-M269 occurs at moderate frequencies among the Iranian-speaking populations of the Caucasus (26%), Azerbaijanis (23%), and Kumyks (19%, Figure 2, Appendix A). Across West Asia, the contribution of R1b is significantly higher for Turkic-speaking populations (20% for Azerbaijanis and Anatolian Turks [20,22,43]) than for Iranian speakers (5%, [20,44,45,46]). Similar R1b-M269 frequencies among the Iranian-speaking populations of the Caucasus and the Turkic-speaking peoples of West Asia may indicate genetic interactions between their ancestral populations. R1b-M73 occurs at observable frequencies among the Karanogais (13%), the populations of the North Caspian steppe (Figure 2, Appendix A), and the Turkic-speaking populations of the West Caucasus (the Karachai and the Balkar, [18,23]).

Haplogroup R1a is represented largely by the R1a-M198(×M458) branch, which is very common among the Karanogais (18%) and Karapapakhs (19%) but occurs at lower frequencies in other populations of the East Caucasus analyzed in this paper (2–11%, Appendix A), except for the Tsez, Kubachi, and Tabasaran. Haplogroup R1a is widespread in Eurasia: except for the R1a-M458 branch, which is typical for the populations of Eastern Europe and rarely found in the populations of the East Caucasus, R1a comprises an astonishing variety of subclades that span a vast geography from India to Scandinavia [47]. Further analysis of R1a-M198(×M458) phylogeography in the local populations of the East Caucasus and the adjacent regions will allow us to identify the sources of migration.

### 3.2. The Main Patterns of Geographic Variation

The frequency distribution maps reveal the patterns of geographic distribution for individual haplogroups that can be summarized for further analysis. Another cartographic method for exploring these patterns is based on the maps of Nei’s genetic distances that aggregate data for all the haplogroups included in the analysis. Every map shows the degree of similarity between the studied populations (Figure 3). The maps of genetic distances expose differences between populations along the entire spectrum of the studied haplogroups, provide a general characteristic of the Y-chromosome gene pool for each population, and highlight regions with the most genetic similarity.

*The Dagestani pattern.* The maps of genetic distances from the Avar-Andi-Dido, Lak, Dargin, and Lezgi linguistic groups representing the Nakh-Dagestanian language family (Figure 3A–K) share a common pattern, the only exception being the Tabasaran people (Figure 3K). This Dagestani pattern is characterized by close genetic distances between the populations (0.01 < đ < 0.11), occurs only in Dagestan, and is not correlated with the language. It is quite distinct although not so intense (0.07 < đ < 0.17) for the Turkic-speaking Kumyks (Figure 3M). The Dagestani pattern shows a deep connection between the gene pools of Dagestan’s populations in terms of Y-chromosome haplogroups. This pattern is largely defined by the presence of haplogroup J1-M267(×P58): its frequency distribution map (Figure 2D) shows its peak frequencies for most Dagestani populations. Within the scope of this study, a genetic similarity is observed between the populations of Dagestan and North Iran, which is determined by the contribution of haplogroups J1 and J2 (Figure 2D,F) and may echo the earliest waves of migration into the East Caucasus.

*The Steppe pattern.* The maps of Nei’s genetic distances from Turkic-speaking populations demonstrate significant differences between these groups (Figure 3L–O). The Kumyk gene pool is close to that of the Caucasian-speaking populations of Dagestan (Figure 3M). The Azerbaijani gene pool (Figure 3N–O) is similar to that of Iranian-speaking peoples of the Caucasus. The only pattern that reflects a very distinct connection to the gene pools of the Eurasian steppe is found among the Karanagais (Figure 3L). This pattern is not characteristic of other populations studied in this paper, except that there is a slight increase in the haplogroup frequency among the Azerbaijani Karapapakhs. It would be fair to call it the “steppe pattern” since it reflects the genetic link to the populations of the Eurasian steppe.

*The Iranian pattern*. There is a genetic similarity between all Azerbaijani populations (0.03 < đ < 0.21, Figure 3N–O) and the Iranian-speaking populations of the Caucasus (Tats, Talysh, Yezidi, Kurds). The contribution of haplogroups J2-M172(×M67, M12) and R1b-M269 to this pattern is the greatest, which can be seen by comparing the maps of their distribution (Figure 2E,H) and the Iranian pattern of genetic distances (Figure 3K,N,O). This pattern manifests on the map of genetic distances from the Tabasaran, who represent the Lezgi branch of the Nakh-Dagestanian language family, although its intensity for the Tabasaran is slightly lower. 

Note. The colors on the maps correspond to the values of genetic distances: the white color is the minimum difference between populations, and the richer the turquoise color, the higher the genetic distances. The color scale (matching colors to values) is shown in the upper right corner of each map.

Legend:

1—Karanogais, 2—Chechen, 3—Kumyks, 4—Tindi, 5—Avars, 6—Dargins, 7—Tsez (Dido) and Hinukh, 8—Laks, 9—Kaitak, 10—Azerbaijanis from Dagestan, 11—Kubachi, 12—Tabasarans, 13—Tsakhur, 14—Rutuls, 15—Lezgins, 16—Karapapakhs, 17—Azerbaijanis (including Karapapakhs) from Azerbaijan, 18—Azerbaijanis (excluding Karapapakhs) from Azerbaijan, 19—Iranian-speaking peoples of the Caucasus (Tats, Talysh, Kurds), 20—Iranian-speaking peoples of the Caucasus (Yezidis).

Overall, the gene pool of the East Caucasus is represented by three main patterns: Dagestani, Iranian, and Steppe. They may correspond to three “layers” of the gene pool formed during different periods in the past. In this hypothesis, the name “Dagestani” shows a connection to the ancient autochthonous Caucasian population that gave rise to a wealth of gene pools (that still retain their unity) due to a powerful genetic drift. The Iranian pattern is associated with an ancient Iranian population descended from Media and earlier waves of migration in the 3rd–2nd millenniums BC. The Steppe pattern is more recent in terms of its origin and its contribution to the gene pool of the East Caucasus; this pattern reflects one of the latest migration waves of Turkic populations.

### 3.3. East Caucasus in Multidimensional Genetic Space

The cartographic analysis of genetic distances identifies the areas of genetic similarity for one population on the one map. In turn, the visual representation of the entire pairwise matrix of Nei’s genetic distances (Table 2) by means of multidimensional scaling reveals the relationship among all studied populations. The constructed genetic space (Figure 4) comprises three different clusters (Figure 4) that reflect geographic but not linguistic patterns. The northeastern vector of Eurasian-steppe influence manifests in the Steppe cluster. The southeastern vector reflects the influence of West Asia and shapes the arbitrary Iranian cluster. The vector of autochthonous Caucasian populations exerts its influence in the Dagestani cluster that unites all Dagestani-speaking peoples. The shortest distance is between the Iranian and Steppe clusters (đ = 0.77). The distance between the Iranian and the Dagestani clusters is slightly longer (đ = 0.93); distances between other pairs of clusters are 1.5–2 times greater (Appendix A). Similar patterns are reproduced in the analysis of principal components (Appendix A, Appendix A).

The Steppe cluster is formed by the Karanogais of Dagestan and the pooled population of Turkic speakers inhabiting the north of the Caspian steppe (Stavropol Trukhmens and Nogais, Astrakhan Tatars, and Nogais). It is characterized by the highest heterogeneity (đ = 0.24, Table 2) and reflects migrations inside the Eurasian steppe, including the medieval migration waves that impacted the gene pools of Nagais and Astrakhan Tatars, the most recent migration of Trukhmens from the Caspian region in the 17th century, etc. Genetic interactions of the past between the populations of the East Caucasus and the Eurasian steppe only show through on the northern border of the region in the gene pool of Dagestani Karanogais and are much weaker for Kumyks and Karapapakhs. 

The Iranian cluster comprises representatives of three language families: the Turkic-speaking Azerbaijanis from Dagestan and Azerbaijan, the Iranian-speaking populations of the Caucasus, and the Lezgic-speaking Tabasaran. This cluster is much denser (đ = 0.14, Table 2) than the Steppe cluster but less consolidated than the Dagestani cluster. The presence of the Tabasaran in this cluster might reflect the legacy of the Tabasaran maisum state in the south of Dagestan, where they lived next to other populations of the cluster, or, alternatively, might indicate the impact of medieval migrations from the Tabaristan province in the north of Iran. The Iranian cluster reflects interactions between the local populations and the populations of West Asia and, therefore, can be called Iranian. The populations of the East Caucasus and West Asia maintained their contacts throughout long periods of history; their contacts occurred across the vast territory that stretches beyond the East Caucasus, so further research is needed to study the sources and history of this component. 

The Dagestani cluster includes all Caucasian-speaking populations of Dagestan, except for the Tabasaran, and the Turkic-speaking Kumyks of Dagestan that are geographically close to each other (the average Nei’s genetic distance between all 11 populations is đ = 0.05, Table 2). The manner in which the populations are distributed within the cluster remotely resembles their geography: the Dargin, Lak, and Tsez populations that live further to the north occupy the top and the left part (“the north-west”) of the plot. Lezgian populations settled further to the south occupy the “south-east” of the plot, i.e., the opposite portion of the cluster. Kumyks’s position on the cluster’s border represents the main layer of their gene pool that connects them to Caucasian-speaking populations and suggests genetic “borrowings” from other Turkic-speaking populations. The periphery of the plot is occupied by the pooled Nakh populations (Chechen and Ingush).

The geographically dense Dagestani component of the East Caucasus gene pool found in most Dagestani populations is largely shaped by the high frequency of haplogroup J1-M267(×P58); its structure will be analyzed below.

### 3.4. Phylogeography of Haplogroup J1-M267(×P58)

Table (Appendix A) shows the results of screening conducted among the populations of the East Caucasus and the neighboring regions. The DNA samples were screened for 31 SNPs that characterize the J1-M267(×P58) lineage. Branch J1-Y3495 dating back to 6.4 ± 0.6 kya is the most prevalent in the region ([https://www.yfull.com/tree/J-Y3476/, accessed on 31 May 2023], Figure 5A, Appendix A). 

Note. The red dots on the map represent the analyzed populations. Areas of grey indicate haplogroup frequencies below 1%, areas of green indicate low frequencies of 1–20%, areas of yellow are the zones of moderate frequencies (20–50%), and areas of red and purple are zones of high frequencies (>50%). 

The region with haplogroup frequency ≥20% in Figure 5B–E is outlined by a line: orange color—for J1-Y3495(×ZS3114,CTS1460), purple—for J1-ZS3114, turquoise—for J1-CTS1460.

Legend:

1—Karanogais, 2—Chechen, 3—Kumyks, 4—Tindi, 5—Avars, 6—Dargins, 7—Tsez (Dido) and Hinukh, 8—Laks, 9—Kaitak, 10—Azerbaijanis from Dagestan, 11—Kubachi, 12—Tabasarans, 13—Tsakhur, 14—Rutuls, 15—Lezgins, 16—Azerbaijanis (including Karapapakhs) from Azerbaijan, 17—Iranian-speaking peoples of the Caucasus (Tats, Talysh, Kurds), 18—Iranian-speaking peoples of the Caucasus (Yezidis).

So far, subhaplogroup J1-Y3495 has not been described in detail in the literature, but its ancestral lineage J1-Z18375 (or phylogenetic equivalent Z1841) is well known [36]. We focused on its phylogenetic equivalent Z1841. For J1-Z1841/Z18375, TMRCA is 8.1 ± 1.0 kya according to YFull [https://www.yfull.com/tree/J-Z1828/, accessed on May 31 2023] and 6.5 ± 1.5 kya according to [36]. This lineage was detected in one of the three representatives of the Kura–Araxes culture (Velikent, Derbent region, Dagestan) that dates back to the Bronze Age (~5000 years ago [48]). This led us to hypothesize its geographic origin in the Caucasus or in the vicinity of the region [36]. Its subbranch J1-Z1842 was found in a man from East Anatolia who lived ~5000 years ago [49]. The fact that the J1-Z1841/Z18375 lineage was common in Anatolia and Levant about 3000 years ago suggests contacts between the speakers of the Hurro-Urartian and the Nakh-Dagestanian languages [36]. 

The only lineage that represents haplogroup J1-Z1841 in the populations of the East Caucasus is J1-Y3495. We were able to identify 17 polymorphic lineages of this haplogroup (Appendix A); frequency distribution maps were constructed for the most common of the identified variants (Figure 5A–D and Appendix A). The maps show that lineages J1-ZS3114 and J1-CTS1460 (~6 kya, Appendix A), as well as J1-Y3495(×ZS3114,CTS1460), occur almost everywhere in the region (Figure 5B–D). There is a cumulative effect: J1-ZS3114 is more common for the speakers of the Dargin, Lak, and Lezgi branches; J1-CTS1460 is typical for the speakers of the Avar-Andi-Dido languages; J1-Y3495(×ZS3114, CTS1460) is widespread among the Dargin speakers and slightly less frequent in the speakers of Lezgi languages (Appendix A). The geographies of these three lineages overlap at a noticeable frequency (over 20%) in Dagestan, suggesting that J1-Y3495 may have originated here (Figure 5E). There is only a slight difference in the average dates of origin between haplogroup J1-Y3495 and its two branches J1-ZS3114 and J1-CTS1460 (300 to 500 years), which indicates population growth in the region in the early Bronze Age (~6 kya), the time of new metalwork technologies, social change, and tribal unions. 

Linguistic data suggests that the Nakh-Dagestanian protolanguage split occurred at the end of the 3rd millennium BC (~5 kya) [50]. This is consistent with our genetic data: older lineages dating back ~6000 years occur in almost all Dagestani populations analyzed in this work and in some Nakh groups (Figure 5, Appendix A), although frequency peaks for J1-CTS1460 and J1-ZS3114 are observed in a number of different linguistic groups. 

Thus, the main portion of the Y gene pool of Dagestan’s populations (carriers of haplogroup J1-M267(×P58) can be traced to the ancestral lineage J1-Y3495. We hypothesize that in the late Copper Age and the early Bronze Age (~6.5 kya), carriers of this lineage were settled in the central part of mountainous Dagestan, where the geographies of the main J1-Y3495 clades overlap (Figure 5E), and were part of the community that spoke the ancestral pan-East-Caucasian language [50]. Other authors [22] came to the same conclusion based on the analysis of three marker sets (genome-wide, Y-chromosome, and mtDNA panels), although the comparison relied on STR markers and only four SNPs. The single founder of the major haplogroup in the gene pool of Dagestan may suggest a reduction of the ancient population in earlier periods when other lineages that co-existed with J1-Y3495 became extinct because of the bottleneck effect. 

The main impact of the Avar-Andi-Dido J1-CTS1460 lineage is observed for its branch J1-Y6916 (5.8 ± 1.1 kya, Appendix A). The gene geography of the most common variants of this lineage (Y6916(×ZS2872), Y61473, ZS2910(×ZS2878, ZS7652), and ZS2878) reveals its geographic center in the areas occupied by Avar-Andi-Dido speakers (Appendix A) and extends to some areas further to the south. The dating of the lineages (4–5 kya, Appendix A) suggests continuous population growth after the lineage split in the diverged population.

## 4. Conclusions

Three components of the East Caucasus gene pool are unequal and may have arisen in different periods of population history.

The Steppe component occurs only in the north of the region among Karanogais and reflects the most recent migration wave of Turkic-speaking nomads from the Eurasian steppe in the Middle Ages. Two other components (Dagestani and Iranian) shape two opposite poles and contribute the most to the East Caucasus gene pool. 

The Dagestani component is largely shaped by lineage J1-Y3495 (6.5 ± 0.6 kya), whose emergence may have been preceded by a population decline followed by a population growth. The analysis of J1-Y3495 phylogeography allowed us to trace its origin to the central part of mountainous Dagestan, to an ancestral population that spoke the pan-East-Caucasian language. The split in the gene pool after ~6 kya can be linked to the population growth, dispersal of communities, and their long-lasting isolation in the mountains. This is confirmed by the split of this haplogroup in Avar-Andi-Dido populations ~4–5 kya.

The Iranian component shows the genetic similarity of Azerbaijanis, the Tabasaran, and the Iranian-speaking populations of the Caucasus to the populations of West Asia that can be linked to a series of ancient migration waves. This pattern results from the contribution of haplogroups J2-M172(×M67, M12) and R1b-M269 that may have been carried into the region by migrations of West Asian tribes, mostly Iranian speakers, throughout its history. Detailed analysis of the origin of this component and its dating will be conducted in the future in a broader genogeographic context.

## Figures and Tables

**Figure 1 genes-14-01780-f001:**
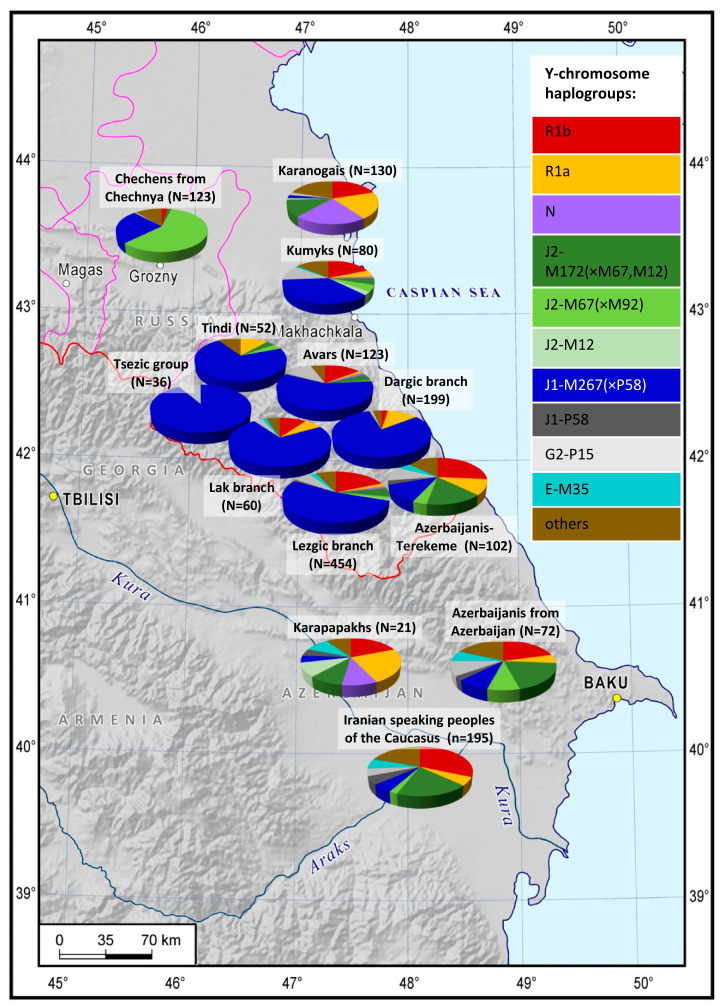
Y-haplogroup spectrum in East Caucasus populations. The colored sections in the pie charts represent requencies of different haplogroups: red—R1b, golden—R1a, purple—N, dark green—J2-M172(×M67,M12), light green—J2-M67(×M92), tea green—J2-M12, blue—J1-M267(×P58), dark grey—J1-P58, light grey—G2-P15, turquoise—E-M35, brown—other.

**Figure 2 genes-14-01780-f002:**
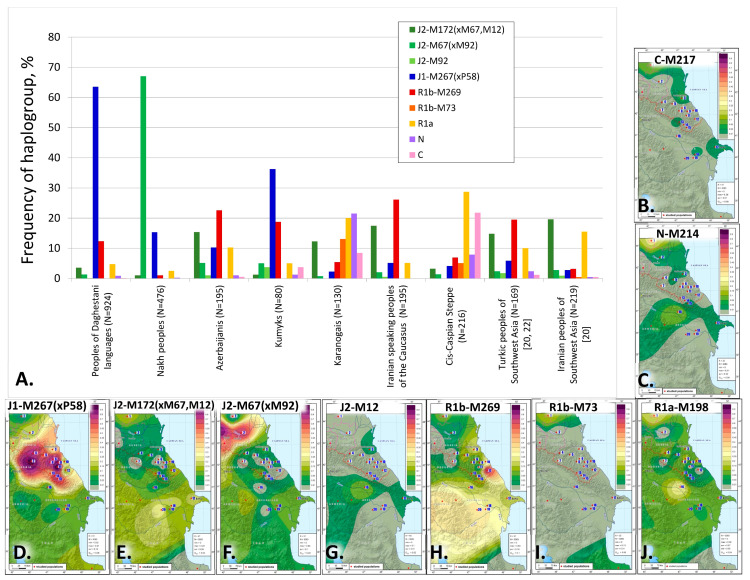
The pattern of the main Y-haplogroups in the East Caucasus and neighboring populations: (**A**) frequency of haplogroups [20,22]; (**B**–**J**) maps of frequency distribution for Y-chromosome haplogroups: (**B**) C-M217; (**C**) N-M214; (**D**) J1-M267(×P58); (**E**) J2-M172(×M67,M12); (**F**) J2-M67(×M92); (**G**) J2-M12; (**H**) R1b-M269; (**I**) R1b-M73; (**J**) R1a-M198.

**Figure 3 genes-14-01780-f003:**
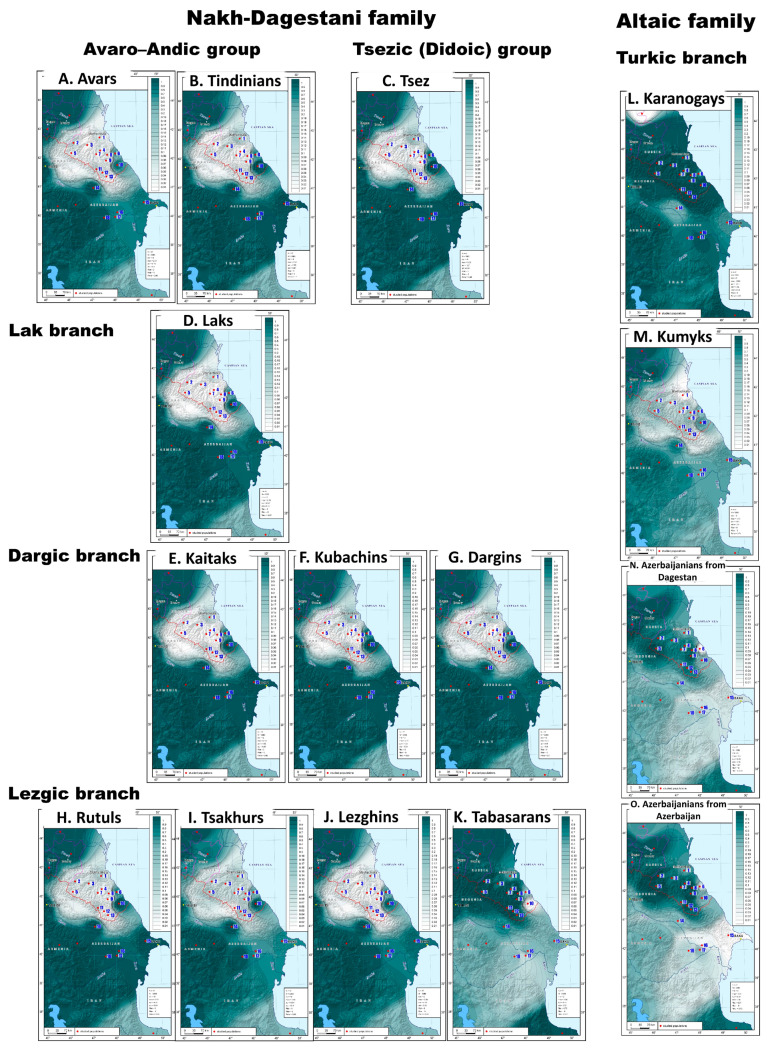
Nei’s genetic distances from 15 Eastern Caucasus populations: (**A**) Avars; (**B**) Tindinians; (**C**) Tsez; (**D**) Laks; (**E**) Kaitaks; (**F**) Kubachins; (**G**) Dargins; (**H**) Rutuls; (**I**) Tsakhurs; (**J**) Lezghins; (**K**) Tabasarans; (**L**) Karanogais; (**M**) Kumyks; (**N**) Azerbaijanians from Dagestan; (**O**) Azerbaijanians from Azerbaijan.

**Figure 4 genes-14-01780-f004:**
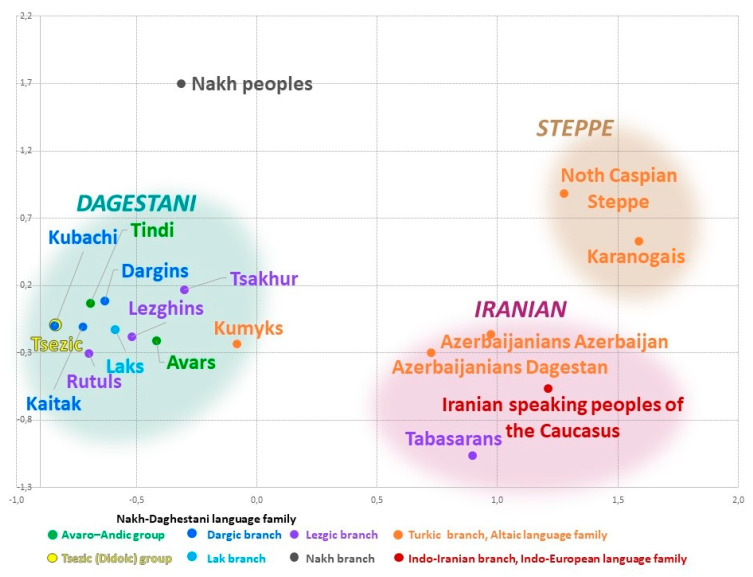
Eastern Caucasus populations on the multidimensional scaling plot (Stress 0.07, Alienation 0.09).

**Figure 5 genes-14-01780-f005:**
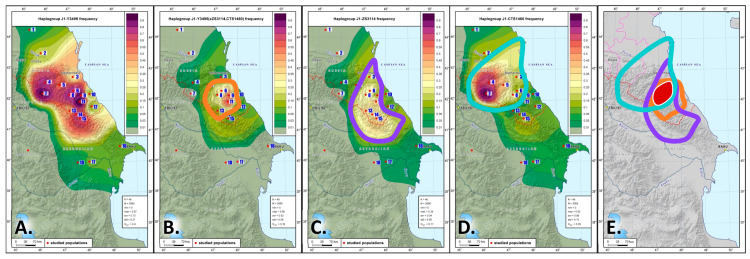
The distribution of Y-haplogroup J1-Y3495 and its main branches in the Eastern Caucasus: (**A**) J1-Y3495; (**B**) J1-Y3495(×ZS3114,CTS1460); (**C**) J1-ZS3114; (**D**) J1-CTS1460; (**E**) area of J1-Y3495 haplogroup origin.

**Table 1 genes-14-01780-t001:** Linguistic and geographic characteristics of the studied populations and samples.

Linguistic Classification		Populations	Region (Country, Province)	N	Data Source
Northeast Caucasian (Nakh-Daghestani) language family	Avaro–Andic group	Avars	Dagestan (Russia)	123	This study, updated from [17]
		Tindi	Dagestan (Russia)	52	This study
	Tsezic (Didoic) group	Tsez (Dido) and Hinukh	Dagestan (Russia)	36	This study
	Dargic branch	Dargins	Dagestan (Russia)	125	This study, updated from [17]
		Kaitak	Dagestan (Russia)	29	Updated from [17]
		Kubachi	Dagestan (Russia)	45	Updated from [17]
	Lak branch	Laks	Dagestan (Russia)	60	This study
	Lezgic branch	Lezghins	Dagestan (Russia)	290	This study, updated from [17]
		Rutuls	Dagestan (Russia)	43	This study
		Tabasarans	Dagestan (Russia)	67	This study
		Tsakhur	Dagestan (Russia)	54	This study
	Nakh branch	Nakh peoples (Ingush and Chechen)	Dagestan, Chechnya, Ingushetia (Russia)	476	This study, updated from [17]
Altaic language family	Turkic branch	Azerbaijanians from Dagestan	Dagestan (Russia)	102	Updated from [23]
		Azerbaijanians (including Karapapakhs) from Azerbaijan	Azerbaijan	93	This study, updated from [23]
		Kumyks	Dagestan (Russia)	80	This study, updated from [23]
		Karanogais	Dagestan (Russia)	130	Updated from [23]
		North Caspian Steppe (Astrakhan Tatars and Nogays, Stavropol Nogays, Stavropol Turkmens)	Astrakhan Region and Stavropol Krai (Russia)	216	This study
Indo-European language family	Iranian group, Indo-Iranian branch	Iranian-speaking peoples of the Caucasus (Tat, Mountain Jews, Talysh, Yezidis, Kurds)	Adygea and Dagestan (Russia); Azerbaijan; Georgia, Armenia	195	This study
3 linguistic families, 7 branches		18 populations	4 countries, 5 regions of Russia	2216	

**Table 2 genes-14-01780-t002:** The pairwise Nei’s genetic distances matrix.

	Avars	Tindi	Tsez (Dido) and Hinukh	Dargins	Kaitak	Kubachi	Laks	Lezghins	Rutuls	Tabasarans	Tsakhur	Nakh Peoples	Karanogais	Kumyks	Azerbaijanians from Dagestan	Azerbaijanians (Including Karapapakhs) from Azerbaijan	North Caspian Steppe	Iranian-speaking peoples of the Caucasus
Avars	0.00	0.06	0.04	0.06	0.03	0.05	0.02	0.02	0.06	0.82	0.08	1.44	1.74	0.07	0.60	0.75	1.61	0.96
Tindi	0.06	0.00	0.02	0.01	0.01	0.02	0.02	0.04	0.05	1.59	0.06	1.28	1.70	0.15	0.87	1.13	1.46	1.65
Tsez (Dido) and Hinukh	0.04	0.02	0.00	0.03	0.01	0.00	0.02	0.03	0.04	1.60	0.10	1.50	2.53	0.16	1.05	1.35	2.10	1.88
Dargins	0.06	0.01	0.03	0.00	0.02	0.03	0.02	0.05	0.05	1.48	0.09	1.45	1.53	0.14	0.83	1.08	1.25	1.56
Kaitak	0.03	0.01	0.01	0.02	0.00	0.01	0.01	0.02	0.03	1.28	0.08	1.49	2.17	0.12	0.86	1.16	1.82	1.53
Kubachi	0.05	0.02	0.00	0.03	0.01	0.00	0.02	0.03	0.04	1.60	0.10	1.51	2.58	0.17	1.05	1.38	2.14	1.91
Laks	0.02	0.02	0.02	0.02	0.01	0.02	0.00	0.01	0.03	1.11	0.08	1.48	1.88	0.09	0.75	1.00	1.57	1.31
Lezghins	0.02	0.04	0.03	0.05	0.02	0.03	0.01	0.00	0.03	0.92	0.08	1.43	1.88	0.07	0.63	0.83	1.68	1.07
Rutuls	0.06	0.05	0.04	0.05	0.03	0.04	0.03	0.03	0.00	1.32	0.11	1.47	2.22	0.12	0.84	1.06	1.70	1.49
Tabasarans	0.82	1.59	1.60	1.48	1.28	1.60	1.11	0.92	1.32	0.00	1.09	2.74	1.35	0.51	0.19	0.26	1.45	0.15
Tsakhur	0.08	0.06	0.10	0.09	0.08	0.10	0.08	0.08	0.11	1.09	0.00	0.89	1.29	0.13	0.54	0.64	1.28	0.92
Nakh peoples	1.44	1.28	1.50	1.45	1.49	1.51	1.48	1.43	1.47	2.74	0.89	0.00	2.59	1.13	1.52	1.18	2.27	2.04
Kumyks	0.07	0.15	0.16	0.14	0.12	0.17	0.09	0.07	0.12	0.51	0.13	1.13	1.32	0.00	0.37	0.49	1.04	0.63
Azerbaijanians from Dagestan	0.60	0.87	1.05	0.83	0.86	1.05	0.75	0.63	0.84	0.19	0.54	1.52	0.67	0.37	0.00	0.08	0.79	0.12
Azerbaijanians (including Karapapakhs) from Azerbaijan	0.75	1.13	1.35	1.08	1.16	1.38	1.00	0.83	1.06	0.26	0.64	1.18	0.59	0.49	0.08	0.00	0.74	0.07
Karanogaiss	1.74	1.70	2.53	1.53	2.17	2.58	1.88	1.88	2.22	1.35	1.29	2.59	0.00	1.32	0.67	0.59	0.24	0.79
North Caspian Steppe	1.61	1.46	2.10	1.25	1.82	2.14	1.57	1.68	1.70	1.45	1.28	2.27	0.24	1.04	0.79	0.74	0.00	1.06
Iranian-speaking peoples of the Caucasus	0.96	1.65	1.88	1.56	1.53	1.91	1.31	1.07	1.49	0.15	0.92	2.04	0.79	0.63	0.12	0.07	1.06	0.00

Note. The fill color of cells in the 1st column and the 1st row represents the language group/branch spoken by the studied populations: Avaro–Andic group—green, Tsezic (Didoic)—yellow, Dargic branch—bright blue, Lak branch—light blue, Lezgic branch—violet, Nakh branch—grey, Turkic branch—orange, Iranian group—red. The fill color of the cells with figures represents genetic distances: white and light colors correspond to the minimal differences between populations; the richer the brown color, the greater the genetic distance.

## Data Availability

Data and materials are available in Appendix A.

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
