# Peer review of "Origins of East Caucasus Gene Pool: Contributions of Autochthonous Bronze Age Populations and Migrations from West Asia Estimated from Y-Chromosome Data"

_genes, 2023, doi:10.3390/genes14091780_

Round 1
Reviewer 1 Report
Dear Authors,
In this study, Agdzhoyan et al. conducted interesting research aiming to understand the origin, and genetic history of the East Caucasus gene pool, using 83 SNPs recovered from uniparental marker Y-chromosome. In this study, the authors generated and used publicly available dataset from a large number of male individuals belong to 18 populations, representing Nakh-Dagestani, Altaic, and Indo-European language families. Using this comprehensive dataset, they identified that the the gene pools of the autochthonous East Caucasus populations were influenced by three genetic components, namely Steppe, Iranian, and Dagestani at different time periods. Each of these genetic components can be phylogeographically distinguished from the others by distinct Y-chromosome haplogroups. The data shows that the haplogroup J1-Y3495 is the most common male associated haplogroup in an autochthonous ancestral population in central Dagestan in the Bronze Age, which further dispersed and diverged to other lineages.
The data presented in this study, complement the available human dataset from the region, and is extremely beneficial to science community to understand the complex pattern of human migration in this region over the past millennia. The research is explained, and written very well, the background information is sufficient to grasp the aim of the study, and associated challenges. The discussion, and conclusion is in line with the results. The presentation of the results should improve in terms of provided information in figures, and tables, as well as quality of the images. Beyond these general comments, below, please find minor comments:
Line 45: … were a [the] center of terraced farming …
Line 52: … economic and social [changes] ….
Line 52-53: with the populations of Southeastern
Line 137: “In summary” instead of “summing up …”
Line 138: ….. or [with] a limited number of Y-chromosome
Line 160: make space between suppl and table 1 “SupplTable 1”
Line 161: please describe the table 1 caption in more details.
Line 182: space correction in “SupplTable 2”.
Line 188: space correction in “SupplTable 3”
Line 192: informative description for table 2 caption is needed – info on the colors, values, language groups etc.
Line 196: ….” is now available offline.” What does that exactly mean?
Line 215: more informative figure legend please.
Line 237: delete the “Note.”
Line 241-246: the resolution of the maps (B-J) should increase, as the legend information cannot be read, and without that the genogeographic map is not informative.
Line 249: …… back [to] 9,000 to - 24, 000 years …
The manuscript is clearly written, however some minot editing of English language is required, such as spacing, commas, the uniformity of verb tenses, etc.
Author Response
Dear Reviewer,
Thank you very much for thorough reviewing of our manuscript. Your comments are very important to improve the article. Please, find our answers bellow.
Line 45: … were a [the] center of terraced farming …
Response 1: Thanks, it was corrected in the text.
Line 52: … economic and social [changes] ….
Response 2: Thanks, it was corrected in the text.
Line 52-53: with the populations of Southeastern
Response 3: Could you please clarify what have to be corrected? The term “Southeastern Europe” is widely used as the name of a region.
Line 137: “In summary” instead of “summing up …”
Response 4: Thanks, it was corrected in the text.
Line 138: ….. or [with] a limited number of Y-chromosome
Response 5: Thanks, it was corrected in the text: “… or consider a limited number of Y-chromosome haplogroups.”
Line 160: make space between suppl and table 1 “SupplTable 1”
Response 6: Thanks, it was corrected in the text.
Line 161: please describe the table 1 caption in more details.
Response 7: Thanks, it was corrected in the text.
Line 182: space correction in “SupplTable 2”.
Response 8: Thanks, it was corrected in the text.
Line 188: space correction in “SupplTable 3”
Response 9: Thanks, it was corrected in the text.
Line 192: informative description for table 2 caption is needed – info on the colors, values, language groups etc.
Response 10: Thanks, it was corrected in the text.
Line 196: ….” is now available offline.” What does that exactly mean?
Response 11: Thanks! We have decided that the phrase ” is now available offline.” is not necessary because all data and it sources are shown in Supplementary Tables.
Line 215: more informative figure legend please.
Response 12: Thanks, it was corrected in the text.
Line 237: delete the “Note.”
Response 13: Thanks, it was corrected in the text.
Line 241-246: the resolution of the maps (B-J) should increase, as the legend information cannot be read, and without that the genogeographic map is not informative.
Response 14: Thanks, it will be corrected for the final images.
Line 249: …… back [to] 9,000 to - 24, 000 years …
Response 15: Thanks, it was corrected in the text.
We hope that you will find our manuscript acceptable in its present form.
Thank you very much for your attention.
Please find the paper with corrections in the attachment.

Reviewer 2 Report
This paper examines Y-chromosome SNP variation within and between 18 populations in the East Caucasus and identifies distinct haplogroups associated with three ancestral components: Dagestani, Steppe, and Iranian. One haplogroup is also shown to have originated in central Dagestan. The introduction provides a useful historical account of this important geographic area in modern human evolution.
comments:
(1) Intra- and inter-populational variation of Y chromosomes allows to infer social structure or migration rates of males among populations. Since the variation of Y chromosomes does not show any correlation with language families, it seems worth examining possible differential patrilocality in these examined populations.
(2) The statistical analysis is based on Nei's genetic distances, but it is necessary to specify which distance is used. Also an appropriate reference is needed (e.g., Nei, 1972).
(3) I am not sure about what is meant by "cartographic analysis" and "genogeographic maps". Brief explanation needs to be described, perhaps in addition to the developer. The sentences from line 378 to 381 are unclear unless what "one-to-many principle" and "one-to-one principle" imply.
(4) line 246 and 327. I believe that "genetic drift" is the most commonly used word rather than "gene drift".
Author Response
Dear Reviewer,
Thank you very much for reviewing our manuscript. Your comments led to improvement of our paper.
- Intra- and inter-populational variation of Y chromosomes allows to infer social structure or migration rates of males among populations. Since the variation of Y chromosomes does not show any correlation with language families, it seems worth examining possible differential patrilocality in these examined populations.
Response 1:
Thanks! The ethnicity of the donors of the studied Y-chromosomes was determined by the ethnicity of the male grandfathers of the analyzed individuals. In other words, the situation was reconstructed for three generations ago when the migration dynamics in the Caucasus was insignificant. Moreover, not only in the male line but all the ancestors of each donor for at least three generations belonged to this population. This indirectly indicates that the individual over a long series of generations can represent the ethnic group and the population. Three generations ago, the peoples of the Caucasus were characterized by strict observance of patrilocality.
- The statistical analysis is based on Nei's genetic distances, but it is necessary to specify which distance is used. Also an appropriate reference is needed (e.g., Nei, 1972).
Response 2: Thanks! We used standard Nei's genetic distance (Nei, 1975 – reference was added) with correction for continuity (2n-1)
The formulas for used Nei's genetic distance D:
D= - ln I
Where I:

plu – frequencies of allele u of locus l, indices 1 and 2 correspond to the Ð opulation1 and the Ð opulation2, respectively,
(2n-1) – correction for continuity
- I am not sure about what is meant by "cartographic analysis" and "genogeographic maps". Brief explanation needs to be described, perhaps in addition to the developer. The sentences from line 378 to 381 are unclear unless what "one-to-many principle" and "one-to-one principle" imply.
Response 3: Thank you for the comment! We have clarified sentences from line 378 to 381. Find please in file with corrections. In this study, "genogeographic maps" means maps of the distribution of the haplogroup frequency and maps of genetic distaces (clarification added to the “methods” section). "Сartographic analysis" is synonymous with “Map analysis”. In this study, mapping techniques were used to explore the geography of genetic traits and patterns.
(4) line 246 and 327. I believe that "genetic drift" is the most commonly used word rather than "gene drift".
Response 4: Thanks, it was corrected in the text.
We hope that you will find our manuscript acceptable in its present form.
Thank you very much for your attention.
Please find the paper file with corrections in the attachment.

Reviewer 3 Report
In this manuscript, the authors have studied some SNP markers on Y-chromosomes among 2,216 diverse populations of East Caucasus and provided a detailed analysis of different haplogroups and their associations with the different populations. The study has been quite intensive as far as different haplogroups are concerned. The authors have done a tremendous job of explaining all the haplogroups. However, there are a few things that could make this study a little better and more interesting for the readers of this journal.
Here are a few comments/suggestions/concerns about the study.
1. What is the significance of only 83 SNP markers? Is this number sufficient? I understand that these may be collected from the previous literature, but some explanations inside the article should have been added.
2. Introduction sections may need more revisions. It mostly has historical facts rather than scientific details about the study.
a. More description about the ~13K Y-SNP markers due to NGS may be needed as if there is any relevance with this study. The SNP markers in this study are a subset of that??
b. “The results of early genetic studies are not so informative and cannot be compared to the data from currently existing marker panels. The Azerbaijani gene pool also remains understudied” – The above sentence needs to be explained why the studies are not so informative. What were the challenges and bottlenecks that this study could fully or partially resolve?
c. The next paragraph suggests some good studies on Azerbaijanis. So it looks a little contradictory.
d. The Introduction only contains the following statement about the study, so more about the study could be better.
"The aim of this study was to conduct a systematic analysis of gene pools of Dagestani and Azerbaijani populations with historically close relations and to reconstruct the genetic history of the East Caucasus using a broad panel of Y-chromosome markers.”
3. It was a bit unclear about what improvements or extra things this study is going to achieve as compared to the previous studies. The authors could make some good explanations in the introduction section.
4. The pairwise genetic distance table needs some explanation about the numbers in each cell (x, yy). The table description can be expanded.
5. The "Method" section could add some scripts and commands if used, link to any public database, or how the offline databases were created.
6. The result in Fig 1 should have been explained in detail in the figure legend. The descriptions of different haplogroups, their significance, and the differences could have been added.
7. Fig3 is hard to understand about the genetic distance (blue, and red should have been explained in legend).
8. Some PCA plots could have been added as the main figure to show the clusters clearly.
9. Finally, how this study is good/bad as compared to NGS-based analysis i.e. genotyping and variant detection using high coverage short or long reads? How accurate is the method of genotyping and SNP detection in this study?
10. The authors could also check the relevant populations in the 1000 genome project dataset to see any overlap as those samples are very well-studied.
There are some grammatical errors and some sentences can be improved.
Author Response
Dear Reviewer,
Thank you very much for your great work and careful review of our article. Your remarks and comments will help us to improve it.
- What is the significance of only 83 SNP markers? Is this number sufficient? I understand that these may be collected from the previous literature, but some explanations inside the article should have been added.
Response 1: Thank you.
“The set of studied 83 SNPs includes two subsets: “Core”-array, 62 SNPs, and J1-M267(xP58) array, 31 SNPs (Suppl. Table 2). “Core”-array covers the main haplogroups of the world Y-chromosome tree, specific to the indigenous populations of Northern Eurasia (the region includes the territories of Russia, the countries of the former USSR and Mongolia). Since the J1-M257(xP58) haplogroup is expressed in the gene pool of the Eastern Caucasus [17, 18], this array was used to study the phylogeography of that haplogroup. “ – these phrases were added to the text.
Both arrays of SNP markers were improved based on the study of the gene pool of the indigenous population of Northern Eurasia by our team (under the supervision of Prof. Elena Balanovska, Prof. Oleg Balanovsky) and partly from published sources.
- Introduction sections may need more revisions. It mostly has historical facts rather than scientific details about the study.
Response 2: Thank you. We tried to add explanations to the text according with paragraphs b—d below.
- More description about the ~13K Y-SNP markers due to NGS may be needed as if there is any relevance with this study. The SNP markers in this study are a subset of that??
Response 2a: The phrase “For example, approximately 13,000 Y-SNPs were discovered using the NGS technology in 2015; of them 66% had been previously unknown. [27]).” is needed only to show a sharp jump in the number of SNP markers known to date and the need to expand the panels of genotyping markers for present studies of Caucasus ethnic groups.
- “The results of early genetic studies are not so informative and cannot be compared to the data from currently existing marker panels. The Azerbaijani gene pool also remains understudied” – The above sentence needs to be explained why the studies are not so informative. What were the challenges and bottlenecks that this study could fully or partially resolve?
Response 2b: Thank you. We tried to add explanations to the text:
So, the Azerbaijani gene pool remains understudied because of using a narrow panel of markers or small samples [9-12], or study populations from neighboring regions - Dagestan or Iran [16, 20-21, 23]. It could be complicated to use these data at the current level of Y-haplogroup resolution and to give a comprehensive description of the gene pool of Azerbaijanis. The results obtained by colleagues on the gene pool of Azerbaijanis are used to consider broader issues than the analysis of the genetic structure of the Eastern Caucasus. However, the absence of a significant Central Asian component in Azerbaijanis and their genetic commonality with the populations of the Middle East and the Caucasus paint a very general portrait of their gene pool.
- The next paragraph suggests some good studies on Azerbaijanis. So it looks a little contradictory.
Response 2c: Thank you. We tried to add explanations to the text – find please in Response 2b.
- The Introduction only contains the following statement about the study, so more about the study could be better.
"The aim of this study was to conduct a systematic analysis of gene pools of Dagestani and Azerbaijani populations with historically close relations and to reconstruct the genetic history of the East Caucasus using a broad panel of Y-chromosome markers.”
Response 2d: Thank you. We tried to add explanations to the text:
Thus, a systematic analysis of the gene pool of Dagestan - one of the most ethnically diverse regions of Russia was carried out quite a long time ago using panels of Y-chromosome markers corresponding to their time [17, 18, 22]. Since a high frequency of the J1 haplogroup was shown in the populations of Dagestan for further analysis of the population history of this region it is important to consider the detailed structure of this haplogroup and the distribution of its branches.
- It was a bit unclear about what improvements or extra things this study is going to achieve as compared to the previous studies. The authors could make some good explanations in the introduction section.
Response 3. We have tried to include explanations in the text (phrases in Responses 2b and 2d above and the phrase below).
The results of this work could be useful for further modeling of the settlement of ancient humans in the Caucasus, Western Asia, and Eurasia as a whole, using an ever-increasing set of dDNA data.
- The pairwise genetic distance table needs some explanation about the numbers in each cell (x, yy). The table description can be expanded.
Response 4: Thanks, it was added.
- The "Method" section could add some scripts and commands if used, link to any public database, or how the offline databases were created.
Response 5:
Thank you. Links to Table 1 and Suppl. Table 1 have been added to the text. The Table 1 contain links to published sources the data from which are used here (these sources are also included in the Y-base database).
The statistical methods used are well known, and the mapping parameters are specified in the materials and methods. All initial data for calculations are given in Suppl. Tables 3 and 6.
- The result in Fig 1 should have been explained in detail in the figure legend. The descriptions of different haplogroups, their significance, and the differences could have been added.
Response 6: Thanks, it was added.
- Fig3 is hard to understand about the genetic distance (blue, and red should have been explained in legend).
Response 7: Thanks, it was added in Note for Fig3.
- Some PCA plots could have been added as the main figure to show the clusters clearly.
Response 8: Thank you.
The multivariate scaling (MDS) plot visualizes the genetic distances calculated over the entire spectrum of haplogroups and was therefore chosen as the main one. The PCA method helps to trace the main vectors of variability within the East Caucasus. The results turned out to be close, and the difference is related to the position of the Tabasarans: on the MDS plot they are inside the Iranian cluster, on the PCA plot – in Dagestani cluster. We emphasized this pattern since it may be important for further studies of the gene pool of populations of the Lezgi language branch.
- Finally, how this study is good/bad as compared to NGS-based analysis i.e. genotyping and variant detection using high coverage short or long reads? How accurate is the method of genotyping and SNP detection in this study?
Response 9:
Thank you! Genotyping method is one of the most accurate for SNP detection - Real-time PCR using microarrays on QuantStudio Real-Time PCR Systems, https://www.thermofisher.com/ru/ru/home/life-science/pcr/real-time-pcr/real-time-pcr-instruments/quantstudio-systems.html.
The second genotyping quality control approach is embedded in the design of the marker panels. Each panel includes markers of different haplogroups or branches of haplogroups. Thus, during the genotyping of the sample results were simultaneously obtained for all SNPs of the panel. This allows us to trace the sample's belonging to its haplogroup and a negative signal in relation to the rest.
- The authors could also check the relevant populations in the 1000 genome project dataset to see any overlap as those samples are very well-studied.
Response 10: Thank you for the recommendations. Unfortunately, from East Caucasus region the 1000 genomes dataset (https://www.internationalgenome.org/data-portal/population) contains information only about 2 male samples from Lezgin and 1 male sample from Chechen populations.
We hope that you will find our manuscript acceptable in its present form.
Thank you very much for your attention.
Please find the article file with corrections (red font color) in the attachment.

Round 2
Reviewer 2 Report
I found the paper much readable and informative, particularly the introduction. The only query is about dDNA in line 156. Is it actually aDNA?
Reviewer 3 Report
The authors have responded to all suggestions.